# Analysis of the Versatility of Multi-Linear Softening Functions Applied in the Simulation of Fracture Behaviour of Fibre-Reinforced Cementitious Materials

**DOI:** 10.3390/ma12223656

**Published:** 2019-11-06

**Authors:** Alejandro Enfedaque, Marcos G. Alberti, Jaime C. Gálvez

**Affiliations:** Departamento de Ingeniería Civil: Construcción, E.T.S de Ingenieros de Caminos, Canales y Puertos, Universidad Politécnica de Madrid, C/Profesor Aranguren, 3, 28040 Madrid, Spain; alejandro.enfedaque@upm.es (A.E.); marcos.garcia@upm.es (M.G.A.)

**Keywords:** cohesive fracture of fibre-reinforced concrete, softening functions, fracture behaviour, glass fibre-reinforced concrete, polyolefin fibre, steel fibres

## Abstract

Fibre-reinforced cementitious materials (FRC) have become an attractive alternative for structural applications. Among such FRC, steel- and polyolefin fibre-reinforced concrete and glass fibre-reinforced concrete are the most used ones. However, in order to exploit the properties of such materials, structural designers need constitutive relations that accurately reproduce FRC fracture behaviour. This contribution analyses the suitability of multilinear softening functions combined with a cohesive crack approach for reproducing the fracture behaviour of the FRC mentioned earlier. The performed implementation accurately simulated fracture behaviour, while being versatile, robust, and efficient from a numerical point-of-view.

## 1. Introduction

All cementitious materials are based on cement being the main binding constituent, which is also responsible for providing some of the most relevant properties, such as their compressive strength and modulus of elasticity. These two properties are highly recommended for construction applications, but some other properties conferred by the cementitious matrix are not as beneficial as the two previous ones. For instance, the flexural strength and the tensile strength of the cementitious materials are limited and consequently might be enhanced if possible. This situation appears in concrete, which boasts a remarkable compressive strength and a tensile strength, which, as a rule of the thumb, can be estimated as a tenth of such value. Thus, when constructing structural elements that are subjected to bending moments, the stresses that appear would crack the material and even fracture it if the tensile strength is surpassed. Obviously, such an event would cause an economic impact on society and might also create a situation where physical damage on humans is inflicted. The traditional solution to such situations has been the use of steel bars placed inside the concrete element section that forms the reinforced concrete. This approach has been used in a wide variety of applications both in civil engineering and architecture. However, in the nineteenth century, the possibility of creating a continuous reinforcement in concrete by adding fibres was invented. From this moment onwards, the use of fibres became an option to be considered, based on the positive effect of the randomly distributed fibres in the mechanical properties of concrete.

If fibres are distributed during the mixing of concrete, their distribution can be considered random and the material manufactured is usually termed as fibre-reinforced concrete (FRC). Conventionally, steel fibres have been the most used ones in concrete structural elements and accordingly the material created has been described as steel fibre-reinforced concrete (SFRC). The usage of such fibres has been extensive and enormously successful as the steel fibres provide not only increments in tensile and flexural strength alike but also in shear strength [1,2]. Moreover, the ductility of the material is greatly enhanced. There are applications of steel fibres, such as industrial pavements and airport runways, shotcrete, and precast elements, etc. [3]. These examples were comprised on the basis of results of the experimental campaigns that assessed tensile behaviour, fatigue performance, or the impact response of SFRC [4,5]. As was shown in such studies, steel fibres confer remarkable mechanical properties upon plain concrete, especially at low strains. Nonetheless, the durability of the concrete elements might be compromised in certain environments, as steel fibres are corrodible.

Macro-polymer fibres with good structural capacity have appeared recently as a possible addition to concrete. Such fibres do not suffer corrosion due to their plastic nature, even when the concrete is placed in chemically harmful environments. Moreover, this type of fibres might reduce the economic impact of steel-price increments on concrete. Recent research has shown that polyolefin fibre-reinforced concrete (PFRC) fulfils the requests established in the recommendations [6,7,8,9] and, therefore, their contribution to the properties of the concrete element might be considered in the structural design [10].

While it is true that most of the applications of fibres in civil engineering and architecture imply their use in structural elements, it should not be overlooked that some cementitious materials with non-structural uses are also relevant, as shown in the published literature [11,12]. This is the case of glass-fibre reinforced cement GRC, which is formed by cement mortar and chopped glass fibres. GRC has been employed in reduced-thickness building elements in which shrinkage might cause cracks. The presence of randomly distributed glass fibres helps to avoid this phenomenon. Furthermore, they do not only enhance the flexural and tensile strength but also provide ductility to a certain extent. The use of GRC has ranged from permanent concrete moulds, communication structures, and façade panels to gutters [13]. Such uses of GRC might imply lowering the cost of the structure, while obtaining a nice final appearance of the structures. Such aspects would be useless if notable mechanical properties of GRC were not achieved.

The structural design and numerical codes consider the fracture behaviour of the fibre-reinforced cementitious material, in order to fully exploit the improvements in the mechanical properties and ductility. In order to reproduce the appearance of cracks and their development in the cementitious matrices, several numerical approaches have been applied. Some models have considered that the focus should be on the overall cracking process without dealing with the single cracks. One of such approaches was termed the smeared-crack approach [14]. These implementations were notoriously successful in those cases where there were no localised cracks and a limited crack opening was detected. Some others have been applied if a detailed study of the appearance and development of cracks was required. Recently, one of these approaches was based on enriching the number of degrees of freedom of the nodes that compose the finite element. One such method, known as the extended finite element method (XFEM), consumed a remarkable amount of computer power although it was able to obtain promising results. Such computational costs derived from the increment of the degrees of freedom that were dedicated to simulate the fracture process. Such an approach was not only used to model the bond between carbon fibre-reinforced polymers (CFRP) and concrete but also to reproduce the compressive behaviour of rubberised concrete or the behaviour of concrete reinforced with steel fibres [15,16,17,18]. Unfortunately, the promising results of XFEM were not transferable to all situations and in some other approaches were not applicable. The augmented finite element method (A-FEM) used a similar approach but without the need of introducing additional degrees of freedom, thus saving computational costs [19]. The cohesive crack model was developed by Hillerborg in the early 1970’s [20] and was first applied to plain concrete. Afterwards, the model was modified in order to consider certain material variations, such as those that might appear in other quasi-brittle materials such as brick masonry [21,22,23]. Additionally, the cohesive crack model was able to reproduce the failure of plain concrete under a combination of bending and shear stresses, which is commonly known as mixed mode, due to the combination of modes I and II. One of the main contributions of this model was the absence of a tracking algorithm which pre-determined the location of the cracks [24].

The cohesive crack model based its success on various factors. Among them, the direct applicability of the mechanical properties obtained in laboratory tests supposed significant advantages of such a model. When applied to plain concrete, the model parameters required were: tensile strength, modulus of elasticity, and fracture energy. Such values could be obtained in laboratory by means of standard codes or using several recommendations [25,26,27,28]. Once such parameters are found, the fracture behaviour of the material requires the proposal of certain softening functions. Although several authors have analysed the applicability of exponential functions to plain concrete with successful results, the use of linear, bi-linear, or multi-linear functions cannot be overlooked. The latter functions have been profusely used due to their simplicity and the accurate simulations of the material behaviour obtained.

This study analyses the changes that should be carried out when choosing the type of multi-linear softening function, in order to accurately reproduce the fracture behaviour of several types of FRC. The numerical simulations were compared with the experimental results. In addition, the changes needed to capture the influence of the fibre dosage, the type of fibres, and the variations of the matrix properties are analysed. Consequently, the experimental results obtained with specimens coming from several formulations of PFRC, GRC, and SFRC are simulated. Finally, all softening functions are examined and the trends and differences outlined.

## 2. Description of the Model

The cracking process in fibre-reinforced cementitious materials is a matter that has been focused on in several studies in the last decades. Using different techniques, the behaviour of such materials have been simulated using, for instance, zero thickness elements or employing inverse analysis [29,30]. By applying such techniques, the fracture behaviour of ultra-high strength fibre-reinforced concrete has been simulated [31,32]. Moreover, if the cohesive crack approach is used, both the PFRC and GRC fracture behaviour are successfully reproduced [7,33,34]. The embedded cohesive crack model implemented is based on a central forces model that is explained below.

The fracture behaviour of the material is introduced by using two parameters. The first one is the fracture energy, obtained by means of laboratory tests. The second one is the shape of the softening function. It should be underlined that several softening functions can be proposed while maintaining the same amount of fracture energy. However, boasting the same fracture energy does not necessarily imply obtaining an accurate fracture behaviour reproduction. Consequently, to reproduce the fracture behaviour, not only the value of the amount of fracture energy is needed to be known, but the appropriate shape of the softening function also needs to be found. Therefore, the shape of the softening function might be considered a property of the FRC that could be influenced by the geometric and mechanical properties of the fibres and the characteristics of the fibre–matrix interface among other factors [35].

The softening function defines the behaviour of the material when the tensile strength is surpassed. The initial instant corresponds to a null crack width, and the inability of the material to sustain any stress determines the critical crack opening. The fracture energy is determined by integrating the area below the stress-crack width curve from a null crack width to the critical crack width, *w_c_*. At this crack width, the stress becomes zero. At any other crack opening *w*, the value of the tensile stress is determined by *f(w)*, as is shown in Equation (1).
(1)$$G_{F}=\int_{0}^{w_{c}}f\left(w\right)dw$$

If the maximum stress reaches the tensile strength (*f_ct_*), the fracture behaviour starts and Equation (2) is confirmed.(2)$$f_{ct}\;=f\left(0\right)$$

The first uses of the cohesive crack models implemented linear, bilinear, or even exponential softening functions in order to capture the cracking process of plain concrete [36]. One such possibility can be seen in Equation (3).
(3)$$\sigma =f_{ct\cdot }e^{\left(-\frac{f_{t}\omega }{G_{F}}\right)}$$
where *f_ct_* is the tensile strength and *G_F_* stands for the specific fracture energy. Using such a function, accurate results were found for plain concrete. Figure 1 shows a sketch of the softening function for the mode I fracture of plain concrete.

However, the shapes of the softening functions proposed for plain concrete are not apt for FRC. In order to minimise the expense of checking the suitability of various possibilities, the concept of inverse analysis was adopted. Inverse analysis is based on adjusting the numerical response of the model to the experimental behaviour by a trial-and-error optimisation implemented in an finite element code [37,38]. As the bilinear softening function merged both the accuracy of the simulation and a low computational expense, the beneficial presence of fibres was simulated by adding linear stretches to the softening function. Consequently, the bilinear softening function was transformed into a tri-linear one and the latter into a multi-linear one if more stretches were added.

The model used in this study was based on the embedded cohesive crack model [21,22], which enabled the numerical simulation of concrete fracture and was extended to FRC. It entailed the assumption of the crack displacement vector **w** to be parallel to the traction vector **t**; with a continuously increasing opening of the crack |**w**|, this relation can be seen in Equation (4).
(4)$$t=\frac{f\left(\left|w\right|\right)}{\left|w\right|}w$$

In order to consider the unloading processes, the cohesive crack unloads to the origin and Equation (4) becomes Equation (5), (see Figure 1).
(5)$$t=\frac{f\left(\left|\tilde{w}\right|\right)}{\left|\tilde{w}\right|}w\;\mathrm{being}\;\tilde{w}=max\left(\left|w\right|\right)$$
where $\tilde{w}$ is the historical maximum magnitude of **w**.

The constitutive relations were implemented in a material subroutine within a FEM code. The commercial code chosen was ABAQUS (ABAQUS version 13, Dassault Systemes, Vélezy-Villacoublay, France) and the implementation was performed by the means of a material user subroutine that used the element geometry recorded in an auxiliary file. The material behaviour was introduced in the program by means of a constitutive relation with different behaviour under tensile and compressive stresses. Under compressive stresses, the material behaved like a linear elastic being its module of elasticity, which has been found in laboratory tests. In addition, no damage under compressive stresses was considered. In the case of tensile stresses, before reaching the tensile strength of the material the behaviour was linear elastic, with the stress–strain relation being governed by the modulus of elasticity. However, once the tensile strength was reached, the behaviour of the material followed the softening function proposed. In the event of unloading, the material moved towards the origin, the zero-strain and zero-stress situation, in a linear manner. If a reloading process occurred, the material was loaded following the same slope defined in the unloading process, until it reached the maximum crack width previously suffered by the material. If the crack continued growing the remaining of the softening function was followed. These characteristics were common to all proposals of the softening functions tested.

Figure 2a shows a random classical finite element determined by a node arrangement. A straight crack is assumed to be embedded in it. As Figure 2b shows, the crack divides the element in the two sub-domains *A*^+^ and *A^−^*. One of the sides of the crack is taken as the reference, which in this case is the corresponding sub-domain *A^−^* with its normal **n** pointing towards the other side and considering it as the positive normal, **w** is defined as the displacement jump across the crack of the opposite side of the crack, with respect to the reference side (see Figure 2b). Following the strong discontinuity approach (SDA), the approximate displacement field within the element could be expressed as follows:(6)$$u\left(x\right)=\underset{a\in A}{\sum }N_{a}\left(x\right)u_{a}+\left[H\left(x\right)-N^{+}\left(x\right)\right]w$$
where *a* is the index of the element node, *N_a_(**x**)* is the shape function for node *a*, **u***_a_* is the corresponding nodal displacement, *H*(***x***) is the Heaviside jump function across the crack plane, which represents a unit step placed along the crack line that can also be defined as the integral of the Dirac’s *δ* function on the crack line [i.e., H(x)=0 for $\;x\in A^{-}$, H(x)=1 for $\;x\in A^{+}$], and $N^{+}\left(x\right)=\underset{a\in A^{+}}{\sum }N_{a}\left(x\right)$.

From the displacement field, the strain tensor can be determined as a continuous part $\varepsilon^{c}$ plus Dirac’s *δ* function on the crack line. The continuous part, which defines the stress field on the element on both sides of the crack, is obtained by the following:(7)$$\varepsilon^{c}\left(x\right)=\varepsilon^{a}\left(x\right)-{\left[b^{+}\left(x\right)⨂w\right]}^{S}$$
where $\varepsilon^{a}$ and $b^{+}$ are given by
(8)$$\varepsilon^{a}\left(x\right)=\underset{a\in A}{\sum }{\left[b_{a}\left(x\right)⨂u_{a}\right]}^{S}\;$$
(9)$$b^{+}=\underset{a\in A^{+}}{\sum }b_{a}\left(x\right)$$
with $b^{+}=\mathrm{grad}\;N_{a}\left(x\right)$. Additionally, the superscript *S* stands for the symmetric part of a tensor and $\varepsilon^{a}\;$ is the apparent strain tensor of the element that was obtained from the nodal displacements.

As has been said before, an assumption was made regarding the bulk material that is not affected by the cracking process, in order to simplify the computations. It was assumed that the material outside the crack behaves isotropically, with a linear-elastic response. The crack displacement vector **w** is obtained at the level of the crack in the constant strain triangle finite element used, considering it as two internal degrees of freedom.

The implementation follows an algorithm similar to plasticity, in order to calculate the stress tensor in the element. If the elasticity of the bulk material is adopted, as was previously mentioned, the stress tensor can be provided by Equation (10). Thus, the stress tensor is expressed as follows:(10)$$\sigma =E:\left[\varepsilon^{a}-{\left(b^{+}⨂w\right)}^{S}\right]$$

In Equation (10) E. stands for the tensor of elastic moduli. However, the displacement of the crack should be obtained before calculating the result of the stress. The jump vector **w** and the traction vector **t** are related by Equation (4), along the cohesive crack. The traction vector is computed locally for obtaining the exact solution as:(11)$$\bar{t}=\sigma \;n$$

For the finite element, however, the approximate tractions and crack jump vectors should be considered. The traction field along the crack line is approximated by a constant traction **t**, in order to simplify the solution. The corresponding equation is obtained by substituting the foregoing expression caused by stress in Equation (10) into Equation (11) and inputting the result into the cohesive crack Equation (5). The resulting condition is as follows:(12)$$\frac{f\left(\left|\tilde{w}\right|\right)}{\left|\tilde{w}\right|}w=\left[E:\varepsilon^{a}\right]\cdot n-\left[E:{\left(b^{+}⨂w\right)}^{S}\right]n$$
which can be rewritten as
(13)$$\frac{f\left(\left|\tilde{w}\right|\right)}{\left|\tilde{w}\right|}w=\left[E:\varepsilon^{a}\right]\cdot n-\left[n\cdot E:b^{+}\right]w$$
or
(14)$$\left[\frac{f\left(\left|\tilde{w}\right|\right)}{\left|\tilde{w}\right|}1+\left[n\cdot E\cdot b^{+}\right]\right]w=\left[E:\varepsilon^{\alpha }\right]\cdot n$$
where **1** is the second-order unit tensor. This equation is solved for **w** by using the Newton–Raphson method, given the nodal displacements (and so $\varepsilon^{a}$) once the crack is formed, with **n** and $b^{+}$, thus, also being obtained. Additional details of the model can be found in [21,22,23].

## 3. Suitability of the Multilinear Softening Functions

The softening functions applied to simulate the fracture process of FRC were chosen according to the material characteristics. Accurate results are found in literature [23] when the bilinear functions were used for simulating the fracture behaviour of plain concrete. However, when fibres were added, several characteristics of the experimental load-deflection curves suggest the introduction of more complex constitutive relations. When selecting the softening function, not only the fibre geometry, but also some other characteristics such as their tensile strength, modulus of elasticity, or even the type of anchorage between the fibres and matrix had to be considered. Consequently, in the case of a frictional bond between the fibres, with a moderate stiffness, a three-stretch softening function was selected. This approach was applied when reproducing the fracture behaviour of materials, such as GRC or PFRC, where the fibres boast a straight shape.

In the case of steel fibres, some other considerations had to be taken into consideration. Steel fibres are approximately between 2.8 times and 20 times stiffer than glass fibres and polymeric fibres, respectively. Moreover, the type of anchorage depends on the shape of the fibres. Nowadays, steel-fibre manufacturers offer a wide variety of shapes, such as straight, sinusoidal, simple-hooked, multiple-hooked, or flat among others. Nevertheless, hooked steel fibres are the most employed. Such fibres offer a two-way anchorage when added to concrete. First of all, there is a remarkable chemical compatibility between steel and the hydrated cement compounds that generates a frictional bond between the matrix and the fibres. Second, the hooks of the fibres create a mechanical grip between both materials. The importance of the mechanical anchorage is much greater that the chemical one and is responsible for most of the load-bearing capacity of the composite material when the width of the cracks is still reduced. The characteristics previously cited were capital for introducing modifications in the softening functions chosen for GRC and PFRC. Therefore, a four-stretch function was chosen. The outlook of the softening functions can be seen in Figure 3.

The implementation of the proposed softening function for GRC and PFRC can be seen in Equation (15). Nonetheless, the values of the parameters that define the geometry of the softening functions vary remarkably between both materials.(15)$$\left\{ \begin{array}{ll} \sigma =f_{ct}+\left(\frac{\sigma_{k}-f_{ct}}{w_{k}}\right)\;w & if\;\;\;\;0<w\leq w_{k}\\ \sigma =\sigma_{k}+\left(\frac{\sigma_{r}-\sigma_{k}}{w_{r}-w_{k}}\right)\left(w\;–w_{k}\right) & if\;\;w_{k}<w\leq w_{r}\\ \sigma =\sigma_{r}+\left(\frac{-\sigma_{r}}{w_{f}-w_{r}}\right)\left(w\;–w_{r}\right) & if\;\;w_{r}<w\leq w_{f}\\ \sigma =0 & if\;\;w>w_{f} \end{array}\right.$$

In the case of the softening function that corresponds to SFRC, the four-stretch function was implemented, as can be seen in Equation (16)
(16)$$\left\{ \begin{array}{ll} \sigma =f_{ct}+\left(\frac{\sigma_{k}-f_{ct}}{w_{k}}\right)\;w & if\;\;\;\;0<w\leq w_{k}\\ \sigma =\sigma_{k}+\left(\frac{\sigma_{r}-\sigma_{k}}{w_{r}-w_{k}}\right)\left(w\;–w_{k}\right) & if\;\;w_{k}<w\leq w_{r}\\ \sigma =\sigma_{r} & if\;\;w_{r}<w\leq w_{t}\\ \sigma =\sigma_{r}+\left(\frac{-\sigma_{r}}{w_{f}-w_{t}}\right)\left(w-w_{t}\right) & if\;\;w_{t}<w\leq w_{f}\\ \sigma =0 & if\;\;w>w_{f} \end{array}\right.$$

At this point, the final stage is to establish the values of the parameters that define the softening functions. In the case of the three-stretch one, the values *k* (*w_k_*, *σ_k_*), *r* (*w_r_*, *σ_r_*), and *f* (*w_f_*, 0) had to be defined. Correspondingly, in the case of the four-stretch function, the values of *k* (*w_k_*, *σ_k_*), *r* (*w_r_*, *σ_r_*), *t* (*w_t_*, *σ_r_*) and *f* (*w_f_*, 0) had to be established. It should be highlighted that the generic expression that could offer a four-stretch function was modified to consider a stretch of constant stress between *k* and *t.*

The methodology used in order to determine the aforementioned parameters is commonly known as inverse analysis. The process can be observed when applied to one formulation of GRC in Figure 4. In the first stage, a proposal of the parameters, *k*, *r* and *f* is made. Such values are implemented in the material subroutine and after the simulation has been carried, out the correspondent numerical curve is obtained. At the second stage, the accuracy of the numerical calculation as well as the amount of fracture energy consumed (*G_f_*) is checked. It should be underlined that the error was not evaluated at each of the parameters that defined the softening function but by considering the similarity of experimental and numerical fracture curves and the amount of fracture energy consumed. If the prediction does not fit either of the cited parameters, a new proposal of *k*, *r* and *f* is assumed. This step is repeated as many times as necessary, in order to obtain an accurate reproduction of the shape of the experimental curves and the value of the fracture energy.

## 4. Materials and Tests

The test specimens for the numerical simulations were produced in previous experimental campaigns. Two types of cementitious matrixes were used:one for steel and polyolefin fibres and a mortar for GRC. In the case of steel and polyolefin macro fibres, a self-compacting concrete was designed. The mix proportioning was previously achieved with the objectives of maintaining self-compactability even after adding the fibres, but also with moderate cement and admixture contents. The aggregate distribution was designed by the maximum dry density criterion and the paste design required 375 kg/m³ of cement. In addition, Sika Viscocrete-5720 admixture with a 1.25% of cement weight and 200 kg/m³ of limestone powder addition were used. The mix proportioning can be observed in Table 1. The tests were performed in accordance with RILEM TC-187 SOC [39]. According to the standard, a notch of a third of the height of the sample was performed in the centre of the sample and the relation between the span and height in the test was set as 3.0. The loading cylinder was placed in the centre of the sample. For every concrete type, three prismatic specimens of dimensions 430 × 100 × 100 mm³ were cast and tested. The simulations were performed with the average curve of each concrete type.

Regarding GRC, three formulations were used with the mix proportions shown in Table 2. The main differences among them was the use of two admixtures called Powerpozz and Metaver. The former is product of pozzolanic nature whereas Metaver is a kaolin that has been thermally treated. The test boards produced were approximately 1200 × 1200 mm² and 10 mm thick. These boards were produced by simultaneous projection of cement mortar and chopped 38 mm-long glass fibres, using the same process that is commonly used in the GRC industry. The volumetric fraction of fibres was 5%. From each type of GRC board, three rectangular 172 × 55 × 10 mm³ specimens were obtained. As in the case of concrete, TC-187-SOC was intended to be applied in the GRC tests. Nevertheless, the magnitudes of the specimens had to be modified as result of the GRC thickness. If the thickness of GRC were increased, the reduced weight of GRC would increase, losing one of the major advantages of the material. However, the rest of suggestions were followed as close as possible. A deeper detail of the production and testing can be found in [33]. As in the concrete tests and according to the standard, the relation between the span and the height of the sample was set as 3.0 and the depth of the central notch was one third of the height. In the case of the GRC samples, the height was 55 mm.

## 5. Results and Discussion

The implementation that has been previously described was employed to simulate several fracture tests. Initially, fracture tests of 100 × 100 × 430 mm³ specimens of self-compacting PFRC with 3, 4.5, 6 and 10 kg/m³ of 60 mm-long polyolefin fibres were reproduced. The experimental plots show the results obtained in at least three tests. In Figure 5, the results of the simulations can be seen. Such curves clearly show that the tri-linear softening function was able to reproduce the fracture characteristics of PFRC with notable precision. The comparison between the experimental results can be seen in Figure 5b. The implementation carried out has shown versatility, robustness and efficiency from a numerical point-of-view. Moreover, as the performed implementation does not require adding degrees of freedom, in contrast to the X-FEM methods, the computational cost of the calculus is reduced so that all simulations performed are finished in a few hours. Additionally, the multilinear approach has been apt when applied to mix-mode (I and II) fracture tests [40]. By changing the points *k*, *r* and *f* that define the softening function, it was possible to simulate all characteristics of the fracture tests, such as the variations of the minimum post-cracking load which changed markedly among the PFRC. Likewise, the maximum experimental post-cracking loads were captured in the numerical curves, together with the slopes of the after-peak loading branch and the after-peak unloading branch. As can be seen in Figure 5, the experimental curves were precisely reproduced.

In order to check the suitability of the tri-linear softening curves when applied to other cementitious material with fibres, the GRC fracture tests were analysed. Three GRC preparations were simulated in accordance with the experimental results. It should be underlined that among them the only difference was the usage of certain chemical products that intended to inhibit the modification of properties that suffer traditional GRC with aging. The traditional formulation was named GRC and the formulation with Powerpozz and Metaver were termed GRC-P and GRC-M, respectively. The tests carried with the mixes of GRC could not be carried out as stated in any recommendation, as there was no standard suitable for this purpose at that time. Similar to the experimental curves of PFRC, the average of at least three such valid tests are plotted in Figure 6. Such results showed a notably low scatter.

The softening functions implemented in the case of the GRC formulations were tri-linear and consequently were defined by three stretches. This approach is similar to the one taken in the case of PFRC. The modifications that were introduced in the parameters that describe the softening functions were capable of reproducing the fracture tests of GRC with noteworthy precision. Such variation in the parameters were able to adapt the simulated fracture behaviour of GRC to the experimental one and reproduce not only the ductility, and the maximum load sustained, but also the unloading process that the materials showed. Based on the softening curves obtained, it can be said that there was no apparent relation observed between the maximum load of the fracture test and the tensile strength of the GRC formulation. Furthermore, the maximum load registered seemed to be related with the slope of the first stretch of the softening function. Such assumption can be performed by contrasting the curves of GRC-P and GRC. Although in such curves it can be seen that GRC-P boasts a higher maximum load, the tensile strength of both materials is the same. Consequently, it might be the greater slope of the unloading branch might be responsible for such different maximum loads.

The most important parameter of the softening functions that define the ductility of the material is the critical crack width, *w_c_*. Nevertheless, as can be seen in Figure 6, *w_c_* is not the only factor that should be evaluated. Although GRC-M and GRC boast the same value of *w_c_*, the maximum crack mouth opening displacement value CMOD varied between both formulations. Consequently, the slope of the last part of the softening curves might have an influence that needs to be considered.

Lastly, the experimental results obtained in the three-point bending fracture tests of an SFRC with a fibre dosage of 26 kg/m³ were simulated. The tests were conducted on 430 × 100 × 100 mm³ specimens. A process similar to the case of the PFRC specimens was followed for both the concrete production and the fracture tests. RILEM-TOC 187 was the recommendation followed. The experimental curves shown in Figure 7 are the average of at least three successful tests, which also scarcely showed any scatter. In this case, the tests were reproduced numerically by using a multilinear softening function (with four stretches). The modification of the tri-linear functions previously cited was based on the outlook of the experimental curve. Such a curve boasted approximately between 0.5 and 1.2 mm of deflection an area that could be identified as a plateau. Consequently, such a feature was added to the tri-linear softening function. In addition, such plateau could be the reflection of the area where the hooks prevent the fibre from being extracted. Therefore, in this area, the fibres behaved as if they were elastically deformed. After this area, the deformation of the hooks and the extraction of the fibres began and consequently the softening curve showed an unloading branch until the load-bearing capacity of the material vanished. The outlook of the softening function employed can be seen in Figure 7. By defining the values of *k*, *r, t* and *f* and using the cited inverse analysis, an accurate reproduction of the experimental behaviour of the SFRC could be reproduced. It was possible to tune not only the peak load of the test but also other features of great relevance, such as the minimum post-peak load, the maximum post-peak load, the shape and load value of the plateau and the unloading process.

Comparing the characteristics of the functions implemented, it could be said that the maximum load registered in the fracture tests is mainly determined by the tensile strength, only if the slope of the first stretch is a relatively large negative value. If such a value is greater, then the maximum load of the fracture test might be influenced by the combination of the tensile strength and the slope. Such comment has been shown to be valid for several materials, types of fibres or even geometries of the specimens tested. The GRC-P maximum load is a clear example of this situation. In the correspondent fracture curves, the material is capable of increasing the total load sustained although the tip of the notch is partially damaged. The simulations were able to reproduce the previously mentioned phenomenon and the elements placed in the whereabouts of the notch tip were damaged before the maximum load was reached.

In order to complete the discussion of the results, Figure 5, Figure 6 and Figure 7 were compared and analysed in more detail. In the figures, it is possible to realise that the fibre type and their shape had a strong influence in the constitutive relations of the composite material. Short straight fibres, such as glass fibres, which have a higher elasticity modulus than concrete, produce increments of the overall fracture energy and the tensile strength, although a softening behaviour is appreciated in the post-cracking branches. The good tensile properties and the mechanical anchorage of steel-hooked fibres, limited the first unloading branch, and the constitutive relation showed a remarkable horizontal plateau. The polyolefin fibres are macro-fibres with an embossed surface and a lower elasticity modulus than the other fibres and the concrete matrix. This is supposed in the constitutive relation by a trilinear post-cracking behaviour, with three turning points. The first is the beginning of a pronounced descent of strength down to a certain opening of the crack (the second turning point) at which the fibres are capable of bearing the stress, when a recharging branch appears and it seems that various mechanisms such as fibre-bridging and fibre-sliding take place at the same time. At a certain crack opening, the constitutive relation reaches the maximum post-cracking strength and starts discharging again. These three main type of behaviour represent most of the mechanisms of fibre-reinforced composite materials. Having said that, this study has shown how multilinear softening branches are a powerful tool, together with a cohesive fracture behaviour, in order to build the constitutive relation of this type of composite materials.

Additionally, this approach was capable of reproducing the effect of the dosage of fibres in the case of PFRC. It should be mentioned that in the case of GRC and SFRC, a more detailed analysis is required in order to claim the suitability of the multilinear softening functions for reproducing formulations with different fibre dosages.

In Table 3, Table 4 and Table 5, the parameters that define the softening functions used in the numerical simulations can be seen. Observing such tables, it can be perceived that the slope of the branch f_ct_-k was greatly influenced, both by the type of fibre and the fibre–matrix interface. It can be seen that the low stiffness of the polyolefin fibres and the bond between such fibres and the concrete matrix led to a linear behaviour until the peak load (see Figure 5). On the contrary, in the case of GRC, all formulations showed a certain loss of linearity of the curve before the peak load was more notable in GRC-M and GRC-P (see Figure 6). In these formulations, the values of w_k_ were remarkably greater than those of PFRC. Regarding the influence of the coordinates of *r*, it can be noticed that in PFRC, the stress that the material is able to sustain at such crack openings is significantly greater than those established in *k*, therefore a reloading took place. Such an event did not appear in any of the GRC formulations, where in all cases σ_r_ was smaller than σ_k_. Therefore, the unloading process, once started, continued until the failure of the material. In the case of SFRC, the multilinear function used required at least five points to define the material softening behaviour.

## 6. Conclusions

Multilinear softening functions were successfully implemented in a commercial finite element code employing a material user subroutine. Using these functions, the fracture tests of PFRC, SFRC and GRC were simulated with notable precision. The numerical processes carried out showed versatility, robustness and efficiency from a numerical point of view.

It is worth noting that this procedure permitted achieving constitutive relations that could serve for the structural design of elements, with three types of fibre-reinforced cementitious materials. That is, this procedure and model could be used to find the softening functions of FRC. This study showed the outstanding possibilities of multi-linear functions and the cohesive crack model, in order to achieve accurate results. Moreover, it is important to clarify, that in order to use this as a predictive model, the physical meaning of the turning points must be found out and related with other material properties or fibre characteristics or dosages.

The shapes of the fracture curves registered in the tests carried out in GRC and PFRC were accurately reproduced. This was made possible by modifying the points that determined the characteristics of the softening curves that were capable of simulating the load regain characteristics of the PFRC curves and the load decrement experimented by the GRC formulations.

The influence of the chemical products added to the GRC formulations and the effect of the dosage of fibres were analysed by modifying the length and slope of the stretches of the softening curves. Such an approach was able to reproduce the typical ductility, while unloading the GRC and the steep load decrement, followed by a load regain and a gradual unloading characteristic of PRFC.

In case of a high slope in the first stretch of the softening curve, the maximum load register in the fracture test was mostly influenced by the tensile strength of the cementitious material. Nonetheless, when such a slope is less steep, the maximum experimental value is influenced by a combination of such a slope and the tensile strength. As far as the ductility is concerned, it is mainly influenced by *w_c_*.

Lastly, a more complicated anchorage system generated an increment of the number of stretches of the softening functions, as detected when comparing the reinforcement of concrete with hooked steel fibres, polyolefin fibres and glass fibres. Subsequently, in the case of steel fibres with more complex geometries, an increment in the number of hooks or the use of sinusoidal fibres might result in multi-linear softening functions with more than four stretches.

## Figures and Tables

**Figure 1 materials-12-03656-f001:**
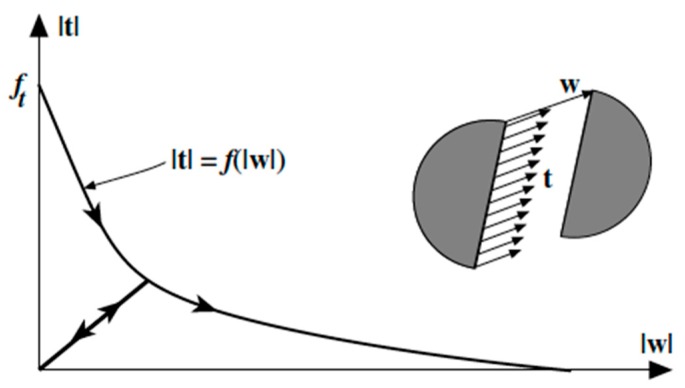
Sketch of the softening curve, with an unloading branch, and the central forces model for the cohesive crack model.

**Figure 2 materials-12-03656-f002:**
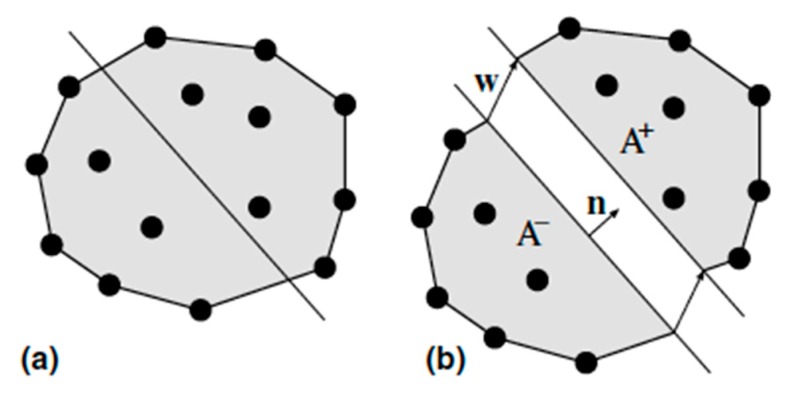
(**a**) Generic element with nodes and crack line and (**b**) displacement jump across the crack line boasting a crack with uniform opening.

**Figure 3 materials-12-03656-f003:**
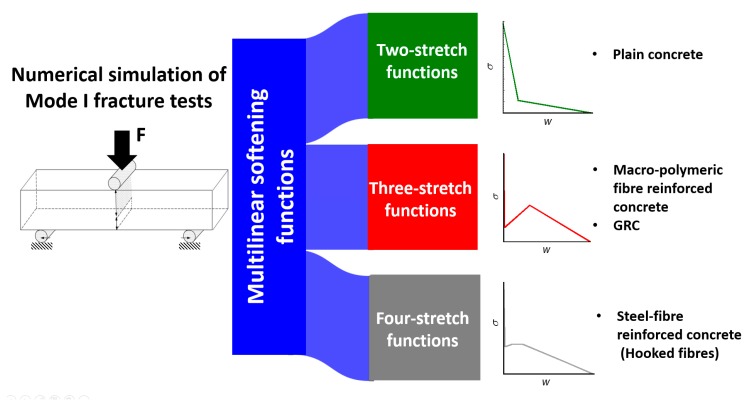
Softening function selection for distinct types of fibres.

**Figure 4 materials-12-03656-f004:**
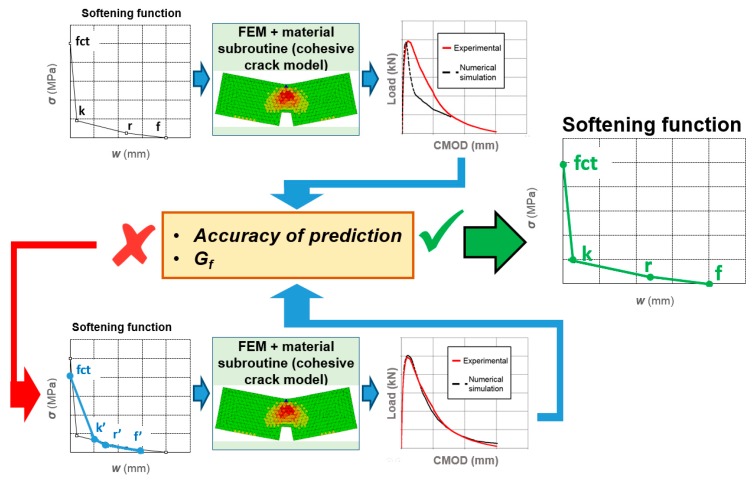
The iterative procedure used to obtain the softening functions.

**Figure 5 materials-12-03656-f005:**
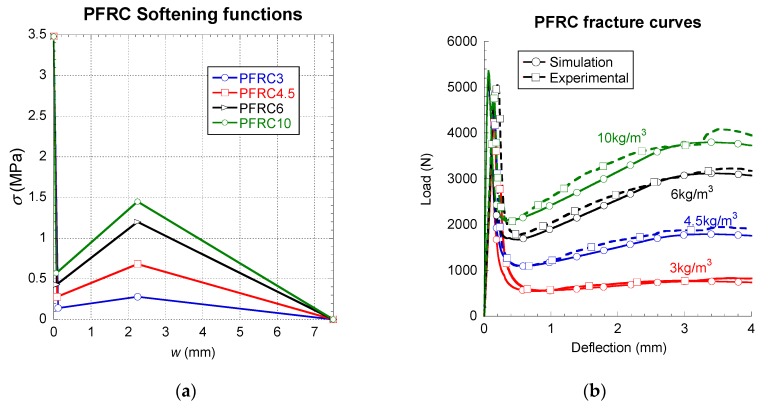
Softening functions (**a**) and comparison among simulated and experimental results (**b**).

**Figure 6 materials-12-03656-f006:**
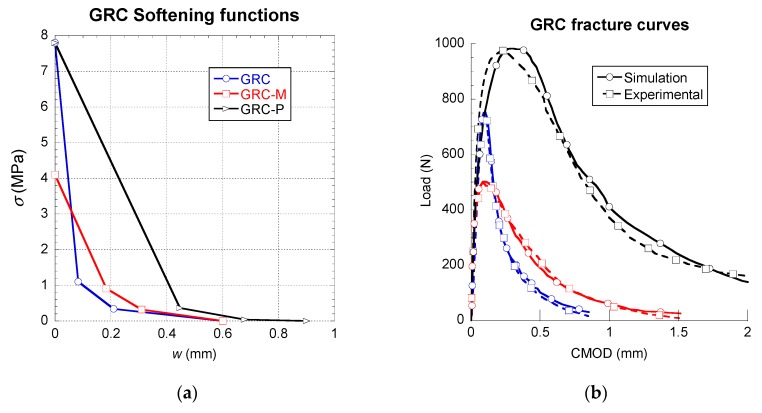
Softening functions (**a**) and comparison of simulated and experimental results (**b**).

**Figure 7 materials-12-03656-f007:**
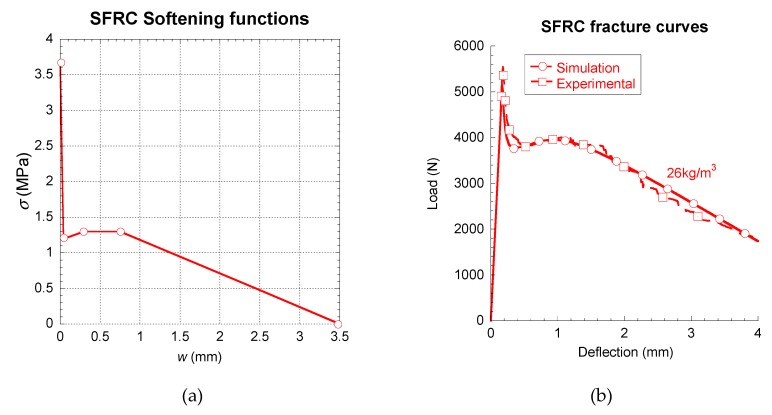
Softening function (**a**) and comparison between the simulated and experimental results (**b**).

**Table 1 materials-12-03656-t001:** Concrete mix proportioning.

Material	SFRC	PFRC3	PFRC4.5	PFRC6	PFRC10
Cement (kg/m^3^)	375	375	375	375	375
Limestone powder (kg/m^3^)	200	200	200	200	200
Water (kg/m^3^)	187.5	187.5	187.5	187.5	187.5
Sand (kg/m^3^)	918	918	918	918	918
Grit (kg/m^3^)	245	245	245	245	245
Gravel (kg/m^3^)	367	367	367	367	367
w/c	0.50	0.50	0.50	0.50	0.50
Steel fibres (kg/m^3^)	26	–	–	–	–
Polyolefin fibres (kg/m^3^)	–	3	4.5	6	10
Superplasticizer (kg/m³)	4.7	4.7	4.7	4.7	4.7

**Table 2 materials-12-03656-t002:** Glass-fibre reinforced cement GRC mix proportioning.

	Cement (kg)	Sand (kg)	Water (kg)	Plasticizer (l)	Addition (kg)
GRC	50	50	17	0.5	-----------
GRC with Metaver^®^ (GRC-M)	50	50	23	0.5	12.5
GRC with Powerpozz^®^ (GRC-P)	50	50	25	0.5	12.5

**Table 3 materials-12-03656-t003:** Parameters of the polyolefin fibre-reinforced concrete (PFRC) softening functions.

	PFRC3		PFRC4,5		PFRC6		PFRC10	
	w (mm)	σ (MPa)	w (mm)	σ (MPa)	w (mm)	σ (MPa)	w (mm)	σ (MPa)
f_ct_	0	3.48	0	3.48	0	3.48	0	3.48
k	0.12	0.14	0.09	0.28	0.08	0.43	0.07	0.57
r	2.25	0.28	2.25	0.68	2.25	1.2	2.25	1.45
f	7.5	0	7.5	0	7.5	0	7.5	0

**Table 4 materials-12-03656-t004:** Parameters of the fibre-reinforced cementitious materials (FRC) softening functions.

	GRC		GRC-M		GRC-P	
	w (mm)	σ (MPa)	w (mm)	σ (MPa)	w (mm)	σ (MPa)
f_ct_	0	7.8	0	4.1	0	7.8
k	0.083	1.1	0.184	0.91	0.447	0.363
r	0.21	0.34	0.31	0.32	0.679	0.04
f	0.6	0	0.6	0	0.9	0

**Table 5 materials-12-03656-t005:** Parameters of steel fibre-reinforced concrete (SFRC) softening function.

	SFRC	
	w (mm)	σ (MPa)
f_ct_	0	3.7
P1	0.04	1.20
P2	0.3	1.3
P3	0.75	1.3
P4	2.45	0.5
p5	3.5	0

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
