# Peer review of "Analysis of the Versatility of Multi-Linear Softening Functions Applied in the Simulation of Fracture Behaviour of Fibre-Reinforced Cementitious Materials"

_materials, 2019, doi:10.3390/ma12223656_

Round 1

Reviewer 1 Report

The manuscript presents a numerical formulation based on a multi-linear softening function for the simulation of the fracture behavior of fiber reinforced cementitious materials. Different fiber types (glass, steel, polyolefin) are considered in the paper, and comparison with experimental results from the relevant literature is discussed to assess the validity of the formulation.

The research work is interesting and the paper is overall well written. However, the following comments and requests of integrations should be taken into account when preparing a revised version of the manuscript.

Introduction, page 1, line 43: in the context of fracture behavior of SFRC, the following paper should be consulted by the authors

Barros, J. A., & Sena-Cruz, J. (2001). Fracture energy of steel fibre reinforced concrete. Journal of Mechanics of Composite Materials and Structures, 8(1), 29-45.

Introduction, page 2, line 61: in the field of fiber reinforced cementitious materials for nonstructural applications, the authors may wish to mention the potential benefits of an innovative lightweight fiber reinforced foamed concrete, presented in

Falliano, D., De Domenico, D., Ricciardi, G., & Gugliandolo, E. (2019). Improving the flexural capacity of extrudable foamed concrete with glass-fiber bi-directional grid reinforcement: An experimental study. Composite Structures, 209, 45-59.

Falliano, D., De Domenico, D., Ricciardi, G., & Gugliandolo, E. (2019). Compressive and flexural strength of fiber-reinforced foamed concrete: Effect of fiber content, curing conditions and dry density. Construction and building materials, 198, 479-493.

This material can be used in the low-density range to realize internal partitions, external infills, and suspended ceilings of buildings, benefitting from the low self-weight, the good mechanical properties conferred by the embedded fibers, and the excellent insulating properties provided by the internal porosity and microstructural features of foamed concrete.

Introduction, page 2, line 79: in the field of recent numerical methods to model discontinuities and cracks in materials, the authors may wish to consider, besides the XFEM, the "augmented-FEM (A-FEM)", presented in

Ling, D., Yang, Q., & Cox, B. (2009). An augmented finite element method for modeling arbitrary discontinuities in composite materials. International journal of fracture, 156(1), 53-73.

The major advantage of the A-FEM in fracture modeling is that it can account for the initiation and propagation of multiple cracks in solids without the need of introducing additional Degrees of Freedom.

The authors should add further comments regarding the computational efficiency of the proposed method in comparison with other approaches. The fiber volumetric ratio should be a key parameter of the multi-linear softening functions, as it directly influences the fracture behavior of the material. However, it seems that the authors have not investigated this parameter in detail. Please add some comments in this regard. It is not clear which parameters are checked in the inverse analysis. Did the author calculate the error in terms of the individual wk, wr, wf values, or did they calculate the error in terms of overall fracture energy (integral error)? There are two tables labeled "Table 1", please check. For reproducibility of the models, the authors should supplement the manuscript with the identified parameters related to the plots in Figs. 5,6,7 for the three materials considered. Moreover, the differences of the identified parameters in terms of fiber type should be discussed more in-depth. Only the multilinear softening functions are shown in Figs. 5-7. In order to emphasize the advantage of the proposed formulation, it is recommended to plot also the bilinear formulation with parameters being identified by inverse analysis (same procedure as the multi-linear formulation). This would give the reader the objective assessment of the importance of multiple parameters (more complicated formulations) compared to simpler approaches. The authors should give more details on the commercial finite element code used. Only mode I fracture tests were simulated by the authors. How do they think the proposed multilinear softening functions can be extended to other types of fracture?

Author Response

The manuscript presents a numerical formulation based on a multi-linear softening function for the simulation of the fracture behavior of fiber reinforced cementitious materials. Different fiber types (glass, steel, polyolefin) are considered in the paper, and comparison with experimental results from the relevant literature is discussed to assess the validity of the formulation.

The research work is interesting and the paper is overall well written. However, the following comments and requests of integrations should be taken into account when preparing a revised version of the manuscript.

Introduction, page 1, line 43: in the context of fracture behavior of SFRC, the following paper should be consulted by the authors

Barros, J. A., & Sena-Cruz, J. (2001). Fracture energy of steel fibre reinforced concrete. Journal of Mechanics of Composite Materials and Structures, 8(1), 29-45.

The authors would like to thank the reviewer for the suggestion. Consequently, the citation has been added to the manuscript.

The usage of such fibres has been extensive and enormously successful as the steel fibres provide not only increments in tensile and flexural strength alike but also in shear strength [1,2]

Introduction, page 2, line 61: in the field of fiber reinforced cementitious materials for nonstructural applications, the authors may wish to mention the potential benefits of an innovative lightweight fiber reinforced foamed concrete, presented in

Falliano, D., De Domenico, D., Ricciardi, G., & Gugliandolo, E. (2019). Improving the flexural capacity of extrudable foamed concrete with glass-fiber bi-directional grid reinforcement: An experimental study. Composite Structures, 209, 45-59.

Falliano, D., De Domenico, D., Ricciardi, G., & Gugliandolo, E. (2019). Compressive and flexural strength of fiber-reinforced foamed concrete: Effect of fiber content, curing conditions and dry density. Construction and building materials, 198, 479-493.

This material can be used in the low-density range to realize internal partitions, external infills, and suspended ceilings of buildings, benefitting from the low self-weight, the good mechanical properties conferred by the embedded fibers, and the excellent insulating properties provided by the internal porosity and microstructural features of foamed concrete.

As in the previous case, the authors agree with the comment and the two citations provided by the reviewer have been incorporated to the text as follows:

While it is true that most of the applications of fibres in civil engineering and architecture imply their use in structural elements, it should not be overlooked that some cementitious materials with non-structural uses are also very relevant as shown in published literature [11, 12].”

Introduction, page 2, line 79: in the field of recent numerical methods to model discontinuities and cracks in materials, the authors may wish to consider, besides the XFEM, the "augmented-FEM (A-FEM)", presented in

Ling, D., Yang, Q., & Cox, B. (2009). An augmented finite element method for modeling arbitrary discontinuities in composite materials. International journal of fracture, 156(1), 53-73.

The major advantage of the A-FEM in fracture modeling is that it can account for the initiation and propagation of multiple cracks in solids without the need of introducing additional Degrees of Freedom.

The authors appreciate the improvement of the introduction suggested by the reviewer and, therefore, a new sentence and the reference have been incorporated to the manuscript. The new wording is the following:

Unfortunately, the promising results of XFEM were not transferable to all situations and in some other approaches were not applicable. The augmented Finite Element Method (A-FEM) used a similar approach but without the need of introducing additional degrees of freedom saving computational costs [19].”

The authors should add further comments regarding the computational efficiency of the proposed method in comparison with other approaches.

As this is a contribution focused in a numerical procedure, the comment performed by the reviewer helps to enhance the quality of the contribution. Consequently, the following wording has been added:

The implementation carried out has shown versatility, robustness and efficiency from a numerical point of view. Moreover, as the performed implementation does not require adding degrees of freedom, in contrast to X-FEM methods, the computational cost of the calculus is reduced so all simulations performed finished in few hours.”

The fiber volumetric ratio should be a key parameter of the multi-linear softening functions, as it directly influences the fracture behavior of the material. However, it seems that the authors have not investigated this parameter in detail. Please add some comments in this regard.

The reviewer has pointed out an interesting issue that is a notable contribution to improve the manuscript. Consequently, the following wording has been added to the manuscript.

Besides, this approach have been capable of reproducing the effect of the dosage of fibres in the case of PFRC. It should be mentioned that in the case of GRC and SFRC a more detailed analysis is required in order to claim the suitability of the multilinear softening functions for reproducing formulations with different fibre dosages.

It is not clear which parameters are checked in the inverse analysis. Did the author calculate the error in terms of the individual wk, wr, wf values, or did they calculate the error in terms of overall fracture energy (integral error)?

The authors consider that the comment of the reviewer clearly improves the overall quality of the paper and thus the following wording has been added:

“The methodology used in order to determine the aforementioned parameters is commonly known as inverse analysis. The process can be observed applied to one formulation of GRC in Figure 4. In the first stage a proposal of the parameters, k, r and f is made. Such values are implemented in the material subroutine and after the simulation has been carried out the correspondent numerical curve is obtained. At the second stage, the accuracy of the numerical calculation is checked as well as the amount of fracture energy consumed (Gf). It should be underlined that the error was not evaluated at each of the parameters that define the softening function but considering the similarity of experimental and numerical fracture curves and the amount of fracture energy consumed.”

There are two tables labeled "Table 1", please check.

The label of Table 2 has been corrected

Table 1. Concrete mix proportioning

Table 2. GRC mix proportioning

For reproducibility of the models, the authors should supplement the manuscript with the identified parameters related to the plots in Figs. 5,6,7 for the three materials considered. Moreover, the differences of the identified parameters in terms of fiber type should be discussed more in-depth.

The authors thank the reviewer for the comment. The authors agree that the inclusion of tables 3, 4 and 5 greatly help the reproducibility of the study. Thus, they have included the tables that follow:

Table 3. Parameters of PFRC softening functions.

PFRC3

PFRC4,5

PFRC6

PFRC10

w (mm)

σ (MPa)

w (mm)

σ (MPa)

w (mm)

σ (MPa)

w (mm)

σ (MPa)

fct

0

3.48

0

3.48

0

3.48

0

3.48

k

0.12

0.14

0.09

0.28

0.08

0.43

0.07

0.57

r

2.25

0.28

2.25

0.68

2.25

1.2

2.25

1.45

f

7.5

0

7.5

0

7.5

0

7.5

0

Table 4. Parameters of FRC softening functions.

GRC

GRC-M

GRC-P

w (mm)

σ (MPa)

w (mm)

σ (MPa)

w (mm)

σ (MPa)

fct

0

7.8

0

4.1

0

7.8

k

0.083

1.1

0.184

0.91

0.447

0.363

r

0.21

0.34

0.31

0.32

0.679

0.04

f

0.6

0

0.6

0

0.9

0

Table 5. Parameters of SFRC softening function.

SFRC

w (mm)

σ (MPa)

fct

0

3.7

P1

0.04

1.20

P2

0.3

1.3

P3

0.75

1.3

P4

2.45

0.5

P5

3.5

0

Moreover, a new and detailed discussion of the values provided have been included in the manuscript.

In tables 3, 4 and 5 the parameters that define the softening functions used in the numerical simulations can be seen. Observing such tables, it can be perceived that the slope of the branch fct-k was greatly influenced both by the type of fibre and the fibre-matrix interface. It can be seen that the low stiffness of the polyolefin fibres and the bond between such fibres and the concrete matrix lead to linear behaviour until the peak load (see Figure 5). On the contrary, in the case of GRC all formulations showed certain loss of linearity of the curve before the peak load being more notable in GRC-M and GRC-P (see Figure 6). In these formulations the values of wk are remarkably greater than those of PFRC. Regarding the influence of the coordinates of r, it can be noticed that in PFRC the stress that the material is able to sustain at such crack openings is significantly greater than those established in k so reloading took place. Such event did not appear in any of the GRC formulations being in all cases σr smaller than σk. Therefore, the unloading process, once started, continued until the failure of the material. In the case of SFRC the multilinear function used required at least 5 points to define the material softening behaviour.

Only the multilinear softening functions are shown in Figs. 5-7. In order to emphasize the advantage of the proposed formulation, it is recommended to plot also the bilinear formulation with parameters being identified by inverse analysis (same procedure as the multi-linear formulation). This would give the reader the objective assessment of the importance of multiple parameters (more complicated formulations) compared to simpler approaches.

The authors would like to emphasise the importance that the bilinear softening function has in modelling the fracture behaviour of plain concrete. As pointed out by the reviewer, there are several studies that have been able to reproduce the fracture behaviour of concrete by means of bilinear curves with a certain degree of accuracy [Guinea, G. V., Planas, J., & Elices, M. (1994). A general bilinear fit for the softening curve of concrete. Materials and structures, 27(2), 99-105.]. However, the same authors in a later contribution pointed out that influence of the very last part of the tail of the P-δ curve is not negligible. In addition, considering such portion is of key importance to obtain accurate results no matter the size of the sample [Elices, M., Guinea, G. V., & Planas, J. (1992). Measurement of the fracture energy using three-point bend tests: Part 3—influence of cutting the P-δ tail. Materials and Structures, 25(6), 327-334.]. In the case of this study, the most important part of the softening curves employed are the ones related with the contribution of the fibres to the material behaviour. Such parts appear for notable crack widths. Moreover, the abscissa of the vertex of the two branches of the bilinear function is defined by the centroid of the area of the fracture energy. In the case of the softening curves of PFRC and SFRC, given that most of the area of the softening function is in the part correspondent to the behaviour of the fibres, the abscissa of such centroid would be in a position that may not be suitable for reproducing accurately the unloading branch that appear in the experimental tests. Consequently, in the opinion of the authors, such curves although scientifically relevant might blur the main issue discussed in this contribution. Moreover, the authors believe that they would not provide relevant information in order to  understand the most relevant advances of the paper. Nonetheless, the authors would have no objection in including it.

The authors should give more details on the commercial finite element code used. Only mode I fracture tests were simulated by the authors. How do they think the proposed multilinear softening functions can be extended to other types of fracture?

The description of the numerical code used has been improved by adding the following wording:

The constitutive relations were implemented in a material subroutine within a FEM code. The commercial code chosen was ABAQUS and the implementation was performed by means of a material user subroutine that used the element geometry recorded in an auxiliary file.”

Moreover, the authors would like to thank the reviewer for the improvement of the manuscript suggested. The following sentences have been added to the manuscript in order to clarify the suitability of multilinear softening curves to other load cases.

Besides, the multilinear approach have been apt when applied to mix-mode (I and II) fracture tests [40]

Reviewer 2 Report

The main aim of this contribution is to analyze the suitability of multilinear softening functions combined with a cohesive crack approach for reproducing the fracture behaviour of the FRC.

This topic is very actual. Methods are clearly written. The results are clearly discussed. 

Author Response

Reviewer 2

The main aim of this contribution is to analyze the suitability of multilinear softening functions combined with a cohesive crack approach for reproducing the fracture behaviour of the FRC.

This topic is very actual. Methods are clearly written. The results are clearly discussed.

The authors would like to thank the reviewer for his/her kind comments .

Reviewer 3 Report

Manuscript ID: materials-620733Type of manuscript: ArticleTitle: Analysis of the versatility of multi-linear softening functions applied to the simulation of the fracture behaviour of fibre reinforced cementitious materials

The article analyses the suitability of multilinear softening functions combined with a cohesive crack approach for reproducing the fracture behaviour of the fibre reinforced cementitious materials, which the reviewer considers justified. The fracture behaviour of the material is introduced by using two parameters. The first one is the fracture energy (fracture curves Load-CMOD), obtained by means of laboratory tests. The second one is the shape of the softening function (the several softening functions can be proposed while maintaining the same amount of fracture energy). For every concrete type (SFRC, PFRC3, PFRC4.5, PFRC6 and PFRC10, constant w/c=0.5), three prismatic specimens of dimensions 430 x 100 x 100 mm³ were cast and tested. The simulations were performed with the average curve of each concrete type.According to the reviewer, both the subject and the proposed approach to solving the task are appropriate. 

Editorial errors

Line 311 “Table 1.” should be “Table 2.”

According to the reviewer, the article is suitable for printing in Materials.

Author Response

Manuscript ID: materials-620733Type of manuscript: ArticleTitle: Analysis of the versatility of multi-linear softening functions applied to the simulation of the fracture behaviour of fibre reinforced cementitious materials

The article analyses the suitability of multilinear softening functions combined with a cohesive crack approach for reproducing the fracture behaviour of the fibre reinforced cementitious materials, which the reviewer considers justified. The fracture behaviour of the material is introduced by using two parameters. The first one is the fracture energy (fracture curves Load-CMOD), obtained by means of laboratory tests. The second one is the shape of the softening function (the several softening functions can be proposed while maintaining the same amount of fracture energy). For every concrete type (SFRC, PFRC3, PFRC4.5, PFRC6 and PFRC10, constant w/c=0.5), three prismatic specimens of dimensions 430 x 100 x 100 mm³ were cast and tested. The simulations were performed with the average curve of each concrete type.According to the reviewer, both the subject and the proposed approach to solving the task are appropriate.

The authors would like to thank the reviewer for his/her kind comments.

Editorial errors

Line 311 “Table 1.” should be “Table 2.”

The editorial change required in line 311 has been introduced.

Table 1. Concrete mix proportioning

Table 2. GRC mix proportioning

According to the reviewer, the article is suitable for printing in Materials.